# Effects of Different Concentrations of *Lactobacillus plantarum* and *Bacillus licheniformis* on Silage Quality, In Vitro Fermentation and Microbial Community of Hybrid *Pennisetum*

**DOI:** 10.3390/ani12141752

**Published:** 2022-07-07

**Authors:** Yanchen Zhu, Haoming Xiong, Zhiying Wen, Hanchen Tian, Yiye Chen, Longfei Wu, Yongqing Guo, Baoli Sun

**Affiliations:** College of Animal Science, South China Agricultural University, Guangzhou 510642, China; scauzyc@163.com (Y.Z.); hmxiongscau@163.com (H.X.); zywenscau@163.com (Z.W.); thcscau@163.com (H.T.); chenyiye1234@126.com (Y.C.); longfeiwu2020@163.com (L.W.); yongqing@scau.edu.cn (Y.G.)

**Keywords:** hybrid *Pennisetum*, *Lactobacillus plantarum*, *Bacillus licheniformis*, silage

## Abstract

**Simple Summary:**

In our study, we reported the effects of *Lactobacillus plantarum* (LP) and *Bacillus licheniformis* (BL) on the quality of hybrid *Pennisetum* (HP) silage. The results of this experiment have a certain reference value for the silage production of hybrid *Pennisetum* and the use of the two additives.

**Abstract:**

The purpose of the experiment was to study the effects of different concentrations of *Lactobacillus plantarum* (LP) and *Bacillus licheniformis* (BL) on the quality of hybrid *Pennisetum* (HP) silage. The experiment consisted of five treatment groups. The control group did not use additives, and the experimental groups were added with LP or BL of 1 × 10^5^ cfu/g fresh weight (FW) and 1 × 10^7^ cfu/g FW, respectively. The results showed that LP and BL could increase the in vitro fermentation gas production and reduce the ammonia nitrogen (AN) content in HP silage. Water-soluble carbohydrates (WSC), lactic acid (LA) content, and gas production in the LP group were positively correlated with LP addition, and acetic acid (AA) was negatively correlated with addition. The content of WSC and LA in the LP7 group was significantly higher than that in the control group (*p* < 0.05), and AA was lower than that in the control group (*p* > 0.05). Dry matter (DM), crude protein (CP), and gas production were negatively correlated with the addition of BL, while acid detergent fiber (ADF) content was positively correlated with the addition of BL. Furthermore, in the above indicators, the BL5 group reached a significant level with the control group (*p* < 0.05). The results of 16sRNA showed that the use of LP and BL could increase the relative abundance of *Lactobacillus* and decrease the relative abundance of *Weissella* in HP silage compared with the control group. In conclusion, LP and BL can significantly improve the quality of HP silage. The LP7 group and the BL5 group have the best silage effect. From the perspective of gas production in in vitro fermentation, the LP7 group had stronger fermentability and higher nutritional value.

## 1. Introduction

The shortage of feed resources and the rising prices of feed lead to the high production cost of the breeding industry, which is one of the most important factors restricting the development of the modern breeding industry [1]. To solve this problem, research and exploration have been carried out in the hope of finding new sources of cheap feed. Hybrid *Pennisetum* (HP) is a triploid hybrid produced by the *Pennisetum americanum* and *P. purpureum* [2]. It has the characteristics of strong adaptability, fast growth rate, high yield, good palatability, high crude protein content, more balanced amino acid content, and rich nutrition; as such, it is a high-quality feed for feeding poultry [3]. Through silage processing, not only can the nutritional value of HP be preserved to the greatest extent, but also the palatability can be improved to a certain extent. However, due to the high buffering capacity (BC), low water-soluble carbohydrate (WSC), and the lack of lactic acid bacteria, HP was difficult to silage [4]. Therefore, foreign additives such as inorganic additives, organic additives, and microbial agents are usually added in the ensilage process to improve the success rate of ensilage.

*Lactobacillus plantarum* (LP) belongs to homogenous fermentation lactic acid bacteria, which is currently one of the most commonly used microbial preparations in silage production. Studies have shown that using LP in the silage process can inhibit the growth of yeast and mold, avoid nutrient loss caused by aerobic degradation of silage, and improve the fermentation quality of silage with the increase of the concentration of LP [5,6,7], probably because it can quickly generate a large amount of lactic acid, reduce environmental pH, inhibit harmful microbial activity, reduce the decomposition of protein and WSC, reduce the loss of nutrients in feed, and significantly improve the success rate of silage and the nutritional value [8,9].

*Bacillus licheniformis* (BL) is a Gram-positive thermophilic bacterium commonly found in soil [10]. It not only has a unique biological oxygen capture mechanism to rapidly create an anaerobic environment but also can produce antibacterial active substances to inhibit the growth and reproduction of pathogenic bacteria [11]. Studies have shown that BL is deemed safe for the intended species, humans who consume products made from animals that were fed treated silage, and the environment [12]. BL has strong activities of protease, lipase, and amylase. It can promote the degradation of nutrients in feed so that animals can absorb and utilize feed more fully. Mara G. et al. found that BL had a strong ability to degrade fiber [13]. Similarly, C. A. Oliveira et al. used BL to treat animal feed, which could significantly improve the digestibility of neutral detergent fiber (NDF) and starch in feed and then improve the feed utilization rate [14]. At present, there are few studies on the application of BL in silage, and its influence on silage needs further study.

The HP was used as the silage material, and silage was prepared by using different concentrations of LP and BL. The study aimed to study the effects of LP and BL on the quality of HP silage and their optimal dosage and to provide a reference for their application in the silage processing industry.

## 2. Materials and Methods

### 2.1. Silage Material and Preparation

The fresh HP was collected from the HP planting base in Meizhou City, Guangdong Province (115.82° E, 24.52° N). The plants were harvested at 2.5–3 m using an automatic grass cutter and cut into 1–2 cm pieces. The DM, CP, NDF and ADF content of HP were 32.80%, 4.41%, 69.60%, 43.16%, respectively. LP and BL were purchased from Guangzhou Yidori Biotechnology Co., LTD (Guangzhou, China). The experiment consisted of 5 treatment groups and ensiled with (1) a control group (C) that did not use additives, (2) 1 × 10^5^ cfu/g fresh weight (FW) LP (LP5) (VTR Bio-Tech Co., Ltd., Zhuhai, China), (3) 1 × 10^7^ cfu/g FW LP (LP7), (4) 1 × 10^5^ cfu/g FW BL (BL5) (VTR Bio-Tech Co., Ltd., Zhuhai, China), and (5) 1 × 10^7^ cfu/g FW BL (BL7). After mixing the additive and HP evenly, take about 200 g into a fermentation bag (20 cm × 30 cm). Three replicates were set in treatment. Sealed with an automatic vacuum compressor, stored at room temperature (25–28 °C), and opened after 35 days of fermentation.

### 2.2. Sample Preparation

Five-gram samples were added to 45 mL of distilled water. After leaching for 24 h in a refrigerator at 4 °C, solids and liquids were separated with four layers of gauze, and the filtrate collected was used to determine fermentation parameters. The pH of silage extract was measured by a pB-10 (Sartorius) pH meter, and the contents of butyric acid (BA), propionic acid (PA), acetic acid (AA), isobutyric acid (IBA), valeric acid (VA) and isovaleric acid (IVA) were analyzed by a high-performance gas chromatograph (Agilent 7890B) [15]. Lactic acid (LA) was detected by the p-hydroxybiphenyl colorimetric method [16]. The phenol-sodium hypochlorite colorimetric method was used to determine ammonia nitrogen (AN) content [17]. To ascertain the nutritional makeup of the remaining samples, they were air-blasted and dried at 65 °C for 48 h to calculate the dry matter (DM). A Kjeldahl nitrogen analyzer (Kjeltec 2300 Auto Analyzer, FOSS Analytical AB, Hilleroed, Denmark) was used to calculate the amount of crude protein (CP) in the sample [18]. The Van Soest method (ANKOM A-200I Fiber analysis apparatus, ANKOM Company, Macedon, NY, USA) was used to assess the amount of neutral detergent fiber (NDF) and acid detergent fiber (ADF) [19]. Hemicellulose (HC) content is equal to NDF content minus ADF content. Anthrone-concentrated sulfuric acid colorimetry was used to estimate the number of water-soluble carbohydrates (WSC) [20].

### 2.3. In Vitro Fermentation

#### 2.3.1. Animals and Experimental Diets

Three healthy Holstein cows with a body weight (566 ± 40) kg and permanent rumen fistula were selected as rumen fluid donors. The experimental cows were fed by a single-pen method, and the diet was prepared according to the recommended nutritional requirements of the Chinese dairy cattle breeding standard. The specific diet composition and nutritional level are shown in Table 1. Cows were fed 2 times a day and drank water freely. The rumen fluid was collected after they adapted to the living environment.

#### 2.3.2. Operating Steps

Rumen fluid from fistulas was collected before morning feeding in each experiment, filtered by 4 layers of gauze, and poured into a thermos flask preheated to 40 °C, which was immediately returned to the laboratory after sealing the cap. Artificial rumen buffer was prepared according to Menke et al. [21], then a rumen fluid:artificial rumen fluid ratio of 1:2 was mixed and put into a water bath at 39 °C, and CO_2_ was continuously introduced until the color of the culture medium changed from pink to colorless. Then, the 200 mg sample was weighed to the front end of the syringe, and 30 mL of mixed culture medium was added to the syringe. Then, incubate them upside down on the water bath shaker which had been preheated to 39 °C. Gas production was recorded for 3, 6, 12, 24, and 48 h in vitro fermentations. After culture, the supernatant was centrifuged, and pH and VFA were determined in the same way as above.

Gas production (mL) = test tube gas production (mL)—blank tube gas production (mL).

### 2.4. Analyses of Bacterial Community

Using the DNeasy PowerSoil Kit, total microbial genomic DNA was recovered from the samples (QIAGEN, Inc., Venlo, The Netherlands). The 16S rRNA V3-V4 sections of genomic DNA were amplified using the Pyrobest DNA Polymerase (DR500A, TaKaRa, Kusatsu, Japan), with the use of the primer pairs 338F and 806R (5′-ACTCCTACGGGAGGCAGCA-3′ and 5′-GACTACHVGGGTATCTAATCC-3′). The effective sequences were obtained after removing the chimeras and low-quality sequences, and they were then grouped into operational taxonomic units (OTUs) using UCLUST with a 97 percent similarity criterion. An OTUs table was created after a representative sequence from each OTU was chosen for further taxonomic categorization using the Basic Local Alignment Search Tool (BLAST). The QIIME and R programs (v3.2.0) were mostly used for sequence data analysis. In detail, α-diversity was analyzed via QIIME. Based on UniFrac distance metrics, beta diversity analysis was carried out to examine the structural variance of microbial communities among samples, and primary coordinate analysis was used to display the results (PCoA). The PICRUSt2 database forecast the roles of microorganisms.

### 2.5. Statistical Analysis

Chemical composition, fermentation parameters, and α-diversity data were evaluated using two-way ANOVA in SPSS 25.0. The data processing was Yij = μ + Ai + Cj + (A × C) ij + εij, where Yij was the dependent variable; μ was the overall mean; Ai represents the additive effect; Cj represents the concentration effect; (A × C) Ij represents the interaction between the additive effect and silage concentration; and εij represents the random residual. Duncan’s method was used for multiple comparisons, and *p* < 0.05 indicated statistical significance.

## 3. Results

### 3.1. Chemical Compositions of HP Silage

Table 2 shows that LP and BL may raise the DM content of HP silage and that DM content exhibited a declining trend as concentration increased. When compared to the control group, the DM content in the BL5 group was considerably greater (*p* < 0.05). Between the LP group and the control group, there was no discernible change in the amount of CP (*p* > 0.05). Additionally, the CP content in the BL5 group was much greater than that in the control group and LP5 group (*p* < 0.05). The WSC content increased with the LP supplemental level, and the LP7 group was significantly higher than the control group (*p* < 0.05). There was no significant difference in NDF content between the two bacteria treatment groups and the control group (*p* > 0.05). Both LP and BL could reduce ADF contents in HP silage. ADF content was considerably reduced in the LP group and BL group compared to the control group (*p* < 0.05). The HC content of the LP and BL groups did not differ significantly from that of the control group (*p* > 0.05).

### 3.2. Fermentation Quality of HP Silage

The effects of different strains and concentrations on the fermentation quality of HP silage are shown in Table 3. There was no statistically significant variation in pH between the experimental and control groups (*p* > 0.05). The AN content in the BL group decreased as concentration increased, and the BL7 group was substantially lower than the control group (*p* < 0.05). The AN content in the LP group was lower than that in the BL group at the same concentration, and the LP5 group was significantly lower than the BL5 group (*p* < 0.05). The LA content in the LP7 group was considerably greater than that in the control and BL7 groups, whereas the AA content was lower in the LP7 group although not significantly (*p* > 0.05). There was no statistically significant difference in LA content between the BL and control groups (*p* > 0.05). PA and BA were not found in any of the groups.

### 3.3. Gas Production of HP Silage

The artificial rumen approach was utilized to assess the impact of various strains on HP silage in vitro fermentation. The results (Table 4) show that, as compared to the control group, using LP and BL in silage can greatly boost in vitro fermentation gas output. At various time points, the gas output of each group was greater than that of the control group. The amount of gas produced varied significantly across treatments. The gas output in the LP group climbed gradually as the additional concentration increased, but the trend in the BL group was exactly the reverse, and the gas production was inversely associated with the added concentration. Gas production was substantially greater in the LP7 group than in the control group (*p* < 0.05). The gas production in the BL5 group was considerably greater than in the control group in the first 24 h (*p* < 0.05), and the gas production in the 48 h was higher than in the control group, but the difference was not significant (*p* > 0.05).

The effects of LP and BL at different concentrations on in vitro fermentation indexes of HP silage were shown in Table 5. Neither LP nor BL had a significant influence on the amounts of TVFA, AA, PA, IBA, BA, IVA, and VA in vitro fermented with HP silage (*p* > 0.05).

### 3.4. Bacterial Community of HP Silage

#### 3.4.1. Bacterial Diversity

The experiment mainly studied the α-diversity. Table 6 shows that when LP concentration increased, all indices in the LP group declined. Both the observed species and Chao1 index were lower in the BL group than in the control group. Furthermore, observed species in the BL5 group were considerably lower than in the control group (*p* < 0.05). The other groups’ indices showed no meaningful difference (*p* > 0.05).

Principal coordinate analysis (PCoA) was performed for β-diversity, and the results are shown in Figure 1. The chart shows that there is a clear separation between the experimental and control groups, as well as a clear separation between the LP and BL groups. Furthermore, there was a clear split between the LP5 and LP7 groups, but no clear differentiation between the BL5 and BL7 groups.

#### 3.4.2. Bacterial Composition

From the level of phylum (Figure 2A), *Firmicutes* were the dominant phylum in all treatment groups. In addition, *Proteobacteria* and *Bacteroidetes* were also found with a relative abundance higher than 1% in each group. The microbial community of different treatment groups changed greatly at the phylum level. The proportion of *Firmicutes* decreases with the increase of LP concentration. The proportion of *Proteobacteria* and *Bacteroidetes* increased with the increase of LP concentration. *Bacteroidetes* in the BL group were higher than that in the control group, the proportion of *Firmicutes* phylum was increased, while the proportion of *Proteobacteria* was decreased. The relative abundance of *Firmicutes* in the BL group was higher than that in the LP group, but *Proteobacteria* and *Bacteroidetes* were lower than that in the LP group.

At the genera level (Figure 2B), the bacteria with high relative abundance in each group were *Lactobacillus* and *Weissella*. In addition, the top 10 genera in relative abundance were *Pediococcus*, *Flavobacterium*, *Leuconostoc*, *Magnetospirillum*, *Aequabacterium*, *Halomnas*, *Thermus,* and *Pseudomonas*. Both LP and BL could increase the relative abundance of *Lactobacillus* and decrease the relative abundance of *Weissella* in HP silage, and *Weissella* in the LP group was lower than that in the BL group. In addition, the relative abundance of *Flavobacterium* and *Magnetospirillum* was positively correlated with the addition of LP, and the relative abundance of *Pediococcus* in the LP7 group was lower than that in the control group.

#### 3.4.3. Predicted Functions and Pathways

Based on the OTU tree and OTU gene information in the Greengene database, the functions of the HP silage microbiota were predicted using the PICRUSt2. The top 20 functions and pathways in abundance were chosen based on the functional annotations, action routes, and abundance information of the samples in the database, and their abundance information in each sample was drawn to the heatmap. It can be seen from Figure 3 that at the K level, the top five KEGG orthologs in relative abundance were ABC-2 type transport system ATP-binding protein, sucrose-6-phosphatase [EC:3.1.3.24], ABC-2 type transport system permease protein, LacI family transcriptional regulator, and probable phosphoglycerate mutase. The top five predicted pathways were the biosynthesis of ansamycins, secondary bile acid biosynthesis, fatty acid biosynthesis, D-glutamine and D-glutamate metabolism, and lysine biosynthesis. The heatmaps of anticipated functions and pathways (Figure 3A,B) suggested that the addition of LP and BL may have influenced the dominating functions and pathways.

## 4. Discussions

Silage is one of the most common methods used in animal husbandry to preserve green fodder. The production principle of silage is to make lactic acid bacteria proliferate in an anaerobic environment, consume oxygen to quickly produce a large amount of lactic acid, and reduce the pH value of the environment, thereby inhibiting the activity of harmful microorganisms, so that feed nutrients can be preserved [22].

During the ensilage process, with the vigorous activities of plant respiration and aerobic microbes, nutrients in the feed are constantly consumed, resulting in the production of water, carbon dioxide, and free ammonia, which leads to the decrease of DM content in silage [23]. LP is a homogenous fermentative lactic acid bacterium that can quickly produce a large amount of lactic acid during storage, reduce the environmental pH, inhibit the activities of harmful microorganisms such as yeast and mold, avoid nutrient loss caused by aerobic degradation of silage, and significantly improve the success rate and nutritional value of silage [6]. This was consistent with the results of this study, and DM content in the LP group was higher than that in the control group. BL can produce anti-active substances that have a unique mechanism of biological oxygen uptake and can inhibit the growth and reproduction of pathogenic bacteria [9]. This may be the reason for the higher DM content in the BL5 group than in the control group. However, the BL has strong protease, lipase, and amylase activity, which can promote the degradation of nutrients in the feed [24,25]. Therefore, the DM in the BL7 group was significantly lower than that in the BL5 group. CP is one of the most important nutrient elements in the feed, the content of which is an important indicator of evaluating the quality of silage. In this experiment, the addition of LP had no significant effect on the CP content of HP silage. Furthermore, the BL5 group was significantly higher than the LP5 group and the control group, which may be because BL can effectively inhibit the degradation of CP by microorganisms in silage. However, with the increase of BL addition, the CP content gradually decreased, which may be caused by the protease contained in BL, and it also explains that high concentrations of BL will deplete nutrients in silage and reduce DM content [13]. The high content of NDF and ADF in the feed will affect the feed intake and digestibility of animals [26]. In this experiment, the use of LP had no significant effect on the NDF content of the HP silage but decreased the ADF content of the silage, which is consistent with the findings of Wang et al. and Zhang et al. [27,28]. Moreover, Mara G et al. also found that *Bacillus licheniformis* had a strong ability to degrade fiber, and its main effect may be to promote the decomposition of ADF [13]. WSC is an energy source for microbial activity in silage, and rapid depletion of WSC is often accompanied by strong microbial activity [29]. Compared with the control group, the WSC content of the LP group was significantly increased, and the LP7 group was significantly higher than the control group, indicating that *Lactobacillus plantarum* can effectively inhibit the colonization of harmful microorganisms, reduce the consumption of WSC, and improve the nutritional value of feed, and the result showed a dose-dependent effect. There was no significant difference in WSC between the BL group and the control group.

pH is one of the important indicators to evaluate the quality of silage. Most harmful bacteria will not survive in acidic environments, so the pH below 4.2 is considered one of the criteria for quality silage [30]. In this experiment, the pH values of the treatment groups were all lower than 4.2, which met the standard of high-quality silage. AN is an index reflecting the decomposition degree of amino acids and proteins in silage. The higher the value is, the greater the decomposition degree of protein is, and the worse the fermentation quality is [31]. Moreover, the absorption and utilization capacity of AN in an animal is far lower than that of true protein, so the higher the content of AN, the lower the nutritional value of the feed. The AN content in experimental groups was much lower than that in the control group, showing that LP and BL can effectively inhibit the degradation of protein feed and improve the feed nutrition value. This may be because they can inhibit the activities of harmful microorganisms such as mold and clostridium in silage and avoid the complete decomposition of feed protein into AN. LP is a homofermentative bacteria that can promote the production of LA, which is consistent with the results of this experiment [32]. The AA content of the LP5 group and BL5 group was significantly higher than that of the control group, which may be caused by insufficient sugar. Lactic acid bacteria (LAB) are generally homofermentative bacteria capable of producing LA. However, when the sugar content of the silage is too low, part of the LAB will turn to heterofermentation, producing LA and an equivalent amount of AA [33]. With the increase of LP and BL supplemental, the homofermentative LAB competitively inhibited the activity of heterofermentative LAB, thereby reducing the AA content.

The silage process is often accompanied by intense microbial activities, and the use of silage additives often has a great influence on the microbial community of silage [34]. Through 16S rRNA high throughput sequencing technology, we can have a deeper understanding of how silage additives improve the quality of silage by affecting the microbial community. Goods refers to the coverage of the sample library, and the higher the value of goods, the higher the reliability of the result. The larger the Chao1 and observed species are, the higher the community richness is. The Simpson index and the Shannon index are used in evaluating species diversity. Pielou’s evenness index characterizes sample evenness [35,36]. Goods’ coverage in each group was higher than 0.99, indicating that the sequencing data coverage was good enough to reflect the silage flora of each treatment group. A large number of studies have shown that *Bacillus licheniformis* has anti-pathogen activity and its fermentation products can inhibit the growth of harmful bacteria such as *Clostridium perfringens* and *Staphylococcus aureus* in vitro [37,38]. This is consistent with the results of this experiment. *Bacillus licheniformis* may inhibit the growth of harmful strains and reduce the abundance of silage colonies through its fermentation products. The β-diversity reflects the differences in bacterial communities among samples. From the PCOA figure, we can see that there is an obvious separation between the experimental groups and the control group, indicating that the addition of different strains has a great impact on the microbial community of HP silage.

The application of bacterial fertilizer can not only change the richness and diversity of bacteria but also change the composition and structure of the bacterial community. At the phylum level, the main phyla in each treatment group were *Firmicutes* and *Proteobacteria*, which was consistent with the results of Chen and Zi et al. [30,39]. *Firmicutes* are Gram-positive bacteria, mainly including spore-producing, non-spore-producing and mycoplasma bacteria, which can degrade macromolecular substances, such as cellulose, starch, protein, etc., and increase the abundance of *Firmicutes* will improve the acid production capacity, resulting in the reduction of pH, and further decompose the ADF [40]. This is consistent with the results of this experiment, *Firmicutes* in BL groups were significantly increased compared with the control group, and pH and ADF were decreased compared with the control group. *Proteobacteria* are Gram-negative bacteria, some of which can undergo deamination reactions to generate ammonia, which slows down the decrease of pH [41]. This may be the reason why the pH of the LP groups is difficult to decrease.

The dominant bacteria genus in each treatment group was *Lactobacillus*, which was consistent with the results of Shah et al. [42]. The dominant bacteria genus on the surface of HP was *Lactobacillus*. *Lactobacillus* is the main bacteria of silage fermentation, belonging to the fermentative bacteria homotype. Its fermentation product lactic acid can quickly reduce the pH of silage, inhibit the growth of fungi, clostridium, and other bacteria, and improve the success rate of silage and fermentation quality. *Weissella* is one of the most common heteromorphic fermentation bacteria in silage, which can produce the same amount of AA as LA. With the accumulation of organic acids and the decrease of pH value, its growth will be inhibited [43]. Studies have shown that LA is positively correlated with lactic acid bacteria and negatively correlated with *Weissella* [44]. It can be seen from the results of this experiment that LP can increase the abundance of *Lactobacillus* and reduce the abundance of *Weissella*, thereby increasing the content of LA, reducing the content of AA, and improving the quality of feed silage. Furthermore, the degree of its effect was positively correlated with the concentration of LP addition. The relative abundance of *Lactobacillus* in the BL group was also higher than that in the control group, and the relative abundance of *Weissella* was lower than that in the control group, but the mechanism of action was different from LP. Adding LP can directly increase the abundance of *Lactobacillus*, while BL can quickly create an anaerobic environment to promote the rapid growth and reproduction of *Lactobacillus* through its unique biological oxygen capture mechanism [45]. At the same time, the antibacterial substances produced by its fermentation can effectively inhibit the growth of mold, clostridium, and other harmful microorganisms in silage, so as to increase the relative abundance of some bacteria in forage silage [46]. *Pediococcus* is one of the microorganisms involved in the early fermentation of silage. The content of *Pediococcus* decreases because LP directly increases the relative abundance of *Lactobacilli*. BL has no significant effect on *Pediococcus*. Studies have shown that *Flavobacterium* belongs to *Bacteroidetes* and can decompose complex polysaccharides such as starch, chitin, pectin, and carboxymethyl cellulose [47]. The content of *Flavobacterium* in the LP groups was significantly higher than that in the control group and BL group, which may be the reason for the higher WSC content in LP groups.

The function and pathway of HP silage were predicted using PICRUSt2. The results showed that among the top five functions, the addition of LP reduced the relative abundance of ABC-2 type transport system ATP-binding protein, sucrose-6-phosphatase [EC:3.1.3.24], ABC-2 type transport system permease protein, and LacI family transcriptional regulator, while the use of BL increased the relative abundance of ABC-2 type transport system ATP-binding protein, sucrose-6-phosphatase [EC:3.1.3.24], ABC-2 type transport system permease protein, LacI family transcriptional regulator and probable phosphoglycerate mutase [EC:5.4.2.12]. Similarly, the performance of LP and BL in the top five pathways is also very different. The use of LP decreased the relative abundance of secondary bile acid biosynthesis, fatty acid biosynthesis, D-glutamine and D-glutamate metabolism, and lysine biosynthesis, whereas the use of BL increased the relative abundance of the top five pathways. These results indicate that both LP and BL have a great influence on the function and pathway of the bacterial community in silage, but their modes of action are quite different, which may be related to their mechanism of action in silage, and further research is needed.

In vitro fermentation gas production is an important indicator for predicting the degree of feed fermentation, which can reflect the digestion and degradation characteristics of ruminants to feed [48]. The amount of gas production is related to the nutritional value of feed, rumen microbial activity, and microbial utilization of feed. The higher the nutritional value of the feed, the stronger the fermentability and the higher the gas production [21]. In this experiment, the gas production of each treatment group at each time point was higher than that of the control group, indicating that both LP and BL can improve the fermentability of silage and improve the nutritional value of feed, which is also consistent with the results of the above silage nutritional indicators. The amount of gas production in the LP group is positively correlated with the amount of LP added, while the trend of gas production in the BL group is just the opposite. In the BL group, only the gas production of the BL5 group was significantly higher than that of the control group in the first 24 h. When the time was increased to 48 h, there was no significant difference in the gas production of the BL group. This may be because the use of BL changed the composition of chemical components in the silage, making it easier to use and ferment, but the impact on total gas production is limited. The gas production in the LP7 group was significantly higher than that in the control group from beginning to end, and the gas production in the LP7 group was the highest among the treatment groups at 48 h, indicating that the effect of LP7 was the best and better than that of the BL5 group.

The pH value in the rumen is one of the important indicators to measure the rumen environment. Calsamiglia et al. [49] indicated that the range of rumen pH should be controlled between 5.5 and 7.5. All treatments in this experiment were between 6.60 and 6.82, which is within the normal range. The VFA content of rumen fluid depends on the source of the fermentation substrate, microbial population, and rumen environment, and is an important indicator for evaluating rumen health and fermentation status [50]. There were no significant differences in the VFA indexes among the test groups, indicating that the use of LP and BL would not affect the growth activity of rumen microorganisms and the health of the rumen.

## 5. Conclusions

The use of additives in the silage process can significantly improve the quality of silage. Both LP and BL can increase the in vitro fermentation gas production and reduce the AN content in the silage. Compared with the control group, the content of WSC was significantly increased in the LP7 group, and the contents of DM and CP were significantly increased in the BL5 group. The addition of LP and BL both increased the relative abundance of *Lactobacillus* sp. and decreased the relative abundance of *Weissella* sp. in silage. In addition, the use of LP can also increase the relative abundance of *Flavobacterium* sp. in silage and decrease the relative abundance of *Pediococcus* sp. In conclusion, both LP and BL can improve the quality of HP silage, and the optimum addition amount of LP is 1 × 10^7^ cfu/g FW, and the optimum addition amount of BL is 1 × 10^5^ cfu/g FW. In terms of gas production by in vitro fermentation, the effect of the LP7 group was better than that of the BL5 group. Therefore, 1 × 10^7^ cfu/g FW LP can be considered an additive in the HP silage process.

## Figures and Tables

**Figure 1 animals-12-01752-f001:**
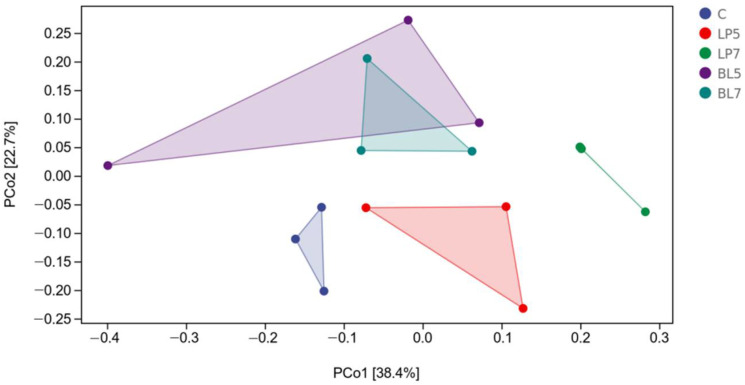
Principal co-ordinates analysis (PCoA) of bacterial communities for HP silage (C, control; LP, *Lactobacillus plantarum*; BL, *Bacillus licheniformis*).

**Figure 2 animals-12-01752-f002:**
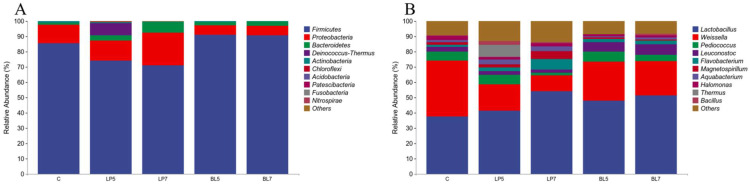
Accumulation map of bacterial communities at the phylum (**A**) and genus (**B**) levels for HP silage (C, control; LP, *Lactobacillus plantarum*; BL, *Bacillus licheniformis*).

**Figure 3 animals-12-01752-f003:**
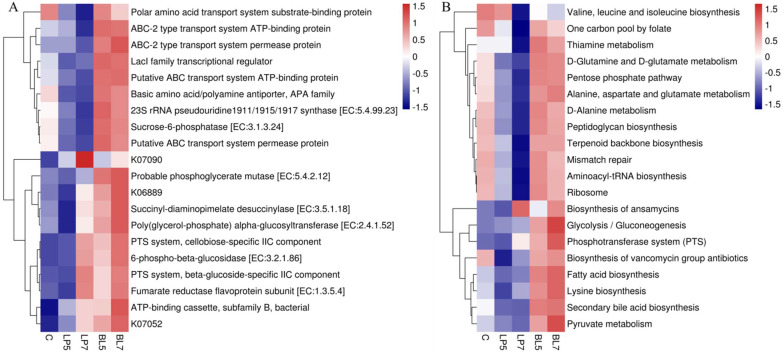
Heatmap of the top 20 predicted functions (**A**) and pathways (**B**) of the bacterial communities analyzed via PICRUSt2 for HP silage (C, control; LP, *Lactobacillus plantarum*; BL, *Bacillus licheniformis*).

**Table 1 animals-12-01752-t001:** Diet composition and nutrient levels (%, DM).

Items	
Flake corn	6.02
Corn	13.27
Soybean meal	15.42
Sunflower meal	0.73
Beet meal	1.68
Whole cottonseed	6.14
Soybean hull	4.29
Puffed soybeans	0.84
Molasses	1.55
^1^ Premix	0.42
CaHPO_4_	0.43
Stone powder	0.87
NaHCO_3_	0.75
MgO	0.22
NaCl	0.28
Corn silage	25.55
Alfalfa	15.26
Oat grass	6.29
Nutrient levels	
CP	17.2
^2^ NEL(Mcal/kg)	6.71
NDF	35.66
ADF	23.65
Ca	0.87
P	0.4
Starch	22.35
NFC	26.61
EE	3.89

^1^ Premix per kg (DM base) contains: VA 1,000,000 IU, VD 392,000 IU, VE 10,080 IU, I 120 mg, Se 66 mg, Co 100 mg, Cu 2000 mg, Zn 18,000 mg, Mn 11,000 mg. ^2^ Net energy for milk production (NEL) is a calculated value.

**Table 2 animals-12-01752-t002:** Effects of different concentrations of LP and BL on nutrients in HP silage.

Items	Treatments	0	10^5^ cfu/g	10^7^ cfu/g	SEM	*p*		
						A	C	A × C
DM	LP	32.40	33.06	32.61	0.086	0.872	0.005	0.475
	BL	32.40 ^b^	33.28 ^a^	32.30 ^b^				
CP (DM%)	LP	4.39	4.25 ^B^	4.32	0.041	0.003	0.162	0.016
	BL	4.39 ^b^	4.92 ^aA^	4.54 ^ab^				
WSC (DM%)	LP	2.58 ^b^	2.87 ^ab^	3.18 ^a^	0.083	0.093	0.236	0.445
	BL	2.58	2.47	2.68				
NDF (DM%)	LP	68.86	67.70	68.32	0.230	0.801	0.172	0.965
	BL	68.86	67.77	68.61				
ADF (DM%)	LP	43.12 ^a^	40.37 ^b^	41.55 ^b^	0.141	0.165	<0.001	0.588
	BL	43.12 ^a^	41.04 ^c^	42.13 ^b^				
HC (DM%)	LP	25.74	27.33	26.77	0.194	0.458	0.051	0.821
	BL	25.74	26.73	26.48				

LP, *Lactobacillus plantarum*; BL, *Bacillus licheniformis*; SEM, standard error of means; A, additives; C, concentration; A × C, the interaction effect of additives and concentration. Different lowercase indicates significant differences in the same row (*p* < 0.05). Different uppercase indicates significant differences in the same column *(p* < 0.05).

**Table 3 animals-12-01752-t003:** Effects of different concentrations of LP and BL on fermentation in HP silage.

Items	Treatments	0	10^5^ cfu/g	10^7^ cfu/g	SEM	*p*		
						A	C	A × C
pH	LP	3.71	3.72	3.71	0.010	0.555	0.111	0.258
	BL	3.71	3.74	3.65				
AN (DM%)	LP	0.91 ^a^	0.42 ^bB^	0.39 ^b^	0.021	0.011	<0.001	0.017
	BL	0.91 ^a^	0.75 ^aA^	0.44 ^b^				
LA (DM%)	LP	6.78 ^b^	6.92 ^b^	8.50 ^aA^	0.229	0.004	0.535	0.029
	BL	6.78	5.51	5.04 ^B^				
AA (mmol/L)	LP	3.58 ^bc^	4.25 ^a^	3.31 ^bB^	0.045	0.334	<0.001	0.028
	BL	3.58 ^b^	4.05 ^a^	3.78 ^abA^				

LP, *Lactobacillus plantarum*; BL, *Bacillus licheniformis*; SEM, standard error of means; A, additives; C, concentration; A × C, the interaction effect of additives and concentration. Different lowercase indicates significant differences in the same row (*p* < 0.05). Different uppercase indicates significant differences in the same column *(p* < 0.05).

**Table 4 animals-12-01752-t004:** Effects of different concentrations of LP and BL on gas production in HP silage (mL).

Items	Treatments	0	10^5^ cfu/g	10^7^ cfu/g	SEM	*p*		
						A	C	A × C
3 h	LP	4.17 ^b^	4.17 ^bB^	7.83 ^a^	0.167	0.011	<0.001	<0.001
	BL	4.17 ^b^	8.17 ^aA^	6.83 ^a^				
6 h	LP	8.67 ^b^	9.00 ^b^	12.00 ^a^	0.236	0.818	0.005	0.009
	BL	8.67 ^b^	11.33 ^a^	10.00 ^ab^				
12 h	LP	13.83 ^b^	14.83 ^b^	17.50 ^a^	0.266	0.684	0.008	0.013
	BL	13.83 ^b^	16.83 ^a^	14.83 ^ab^				
24 h	LP	20.83 ^b^	22.17 ^b^	25.17 ^aA^	0.342	0.635	0.013	0.028
	BL	20.83 ^b^	24.50 ^a^	21.83 ^abB^				
48 h	LP	29.50 ^b^	31.17 ^ab^	34.50 ^aA^	0.487	0.277	0.090	0.064
	BL	29.50	32.50	29.83 ^B^				

LP, *Lactobacillus plantarum*; BL, *Bacillus licheniformis*; SEM, standard error of means; A, additives; C, concentration; A × C, the interaction effect of additives and concentration. Different lowercase indicates significant differences in the same row (*p* < 0.05). Different uppercase indicates significant differences in the same column (*p* < 0.05).

**Table 5 animals-12-01752-t005:** Effects of different concentrations of LP and BL on in vitro fermentation in HP silage.

Items	Treatments	0	10^5^ cfu/g	10^7^ cfu/g	SEM	*p*		
						A	C	A × C
pH	LP	6.76	6.74	6.74	0.002	1.000	0.041	0.133
	BL	6.76 ^a^	6.75 ^ab^	6.74 ^b^				
TVFA (mmol/L)	LP	76.13	79.02	78.61	0.735	0.357	0.573	0.798
	BL	76.13	76.97	76.44				
AA (mmol/L)	LP	46.17	48.39	47.96	0.483	0.268	0.513	0.720
	BL	46.17	46.71	46.27				
PA (mmol/L)	LP	16.87	17.37	17.38	0.150	0.372	0.646	0.810
	BL	16.87	16.96	16.96				
IBA (mmol/L)	LP	1.31	1.32	1.35	0.012	0.970	0.623	0.536
	BL	1.31	1.35	1.32				
BA (mmol/L)	LP	7.71	7.87	7.90	0.067	0.942	0.546	0.980
	BL	7.71	7.88	7.85				
IVA (mmol/L)	LP	2.80	2.81	2.79	0.022	0.887	0.876	0.993
	BL	2.80	2.80	2.78				
VA (mmol/L)	LP	1.26	1.17	1.23	0.011	0.671	0.764	0.792
	BL	1.26	1.26	1.26				

LP, *Lactobacillus plantarum*; BL, *Bacillus licheniformis*; SEM, standard error of means; A, additives; C, concentration; A × C, the interaction effect of additives and concentration. Different lowercase indicates significant differences in the same row (*p* < 0.05).

**Table 6 animals-12-01752-t006:** Effects of different concentrations of LP and BL on α-diversity in HP silage.

Items	Treatments	0	10^5^ cfu/g	10^7^ cfu/g	SEM	*p*		
						A	C	A × C
Observed species	LP	722.23	1386.67 ^A^	806.33	55.49	0.033	0.355	0.077
	BL	722.23 ^a^	515.93 ^bB^	655.97 ^ab^				
Chao1 index	LP	863.99	1703.77	1024.28	70.95	0.028	0.417	0.079
	BL	863.99 ^ab^	587.00 ^b^	734.00 ^ab^				
Shannon index	LP	4.34 ^b^	5.27 ^a^	4.25 ^b^	0.08	0.358	0.048	0.052
	BL	4.34	4.47	4.57				
Simpson index	LP	0.83 ^ab^	0.86 ^a^	0.81 ^b^	0.01	0.489	0.522	0.434
	BL	0.83	0.81	0.82				
Pielou evenness	LP	0.46 ^b^	0.51 ^a^	0.44 ^b^	0.01	0.429	0.084	0.257
	BL	0.46	0.50	0.49				

LP, *Lactobacillus plantarum*; BL, *Bacillus licheniformis*; SEM, standard error of means; A, additives; C, concentration; A × C, the interaction effect of additives and concentration. Different lowercase indicates significant differences in the same row (*p* < 0.05). Different uppercase indicates significant differences in the same column (*p* < 0.05).

## Data Availability

The data presented in this study are available on request from the corresponding author.

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
