# Peer review of "Effects of Different Concentrations of Lactobacillus plantarum and Bacillus licheniformis on Silage Quality, In Vitro Fermentation and Microbial Community of Hybrid Pennisetum"

_animals, 2022, doi:10.3390/ani12141752_

Round 1
Reviewer 1 Report
Summary/Abstract:
unclear what is meant by "theoretical reference" - line 10
in vitro - italicized throughout
16S results confirmatory bur not surprising - LP in yields LP out - line 25 and Figure 2B
Body of manuscript
While the experimental question in the article has merit and addresses an agricultural impactful issue around improving silage quality for animal feed, the article was difficult to read and follow. While there are some minor English language errors the primary issue this article has in terms of readability and accessibilty to a braod audience is the use of abbreivations for products, microbes, etc. without any definition of the abbreviations for the reader. Examples include, DM, AN, HP - while others were defined in the text - for example LP - Lactobacillus plantarum or WSC - water-soluble carbohydrate. Thiis made the article very difficult to read and follow the authors conclusions relative to the expreimental results presented and the approach taken in the methodology section.
Two suggestions that could improve readability and accessibility would be to reformat with results and discussion embedded together along with definitions of all abbreviations.
Minor suggestions include modification of table 3 - if products are undetected that can be stated in text and is not needed in table
in vitro - italicized throughout
Dicussion and Conclusions - tie the reults and offer suggestion for improved silage processing.
Author Response
Thank you for your suggestion. Please see the attachment.

Reviewer 2 Report
The presented study is relevant. The introduction section it is well-written. The methods used were adequate and clearly described. The results are adequately presented and well discussed. As presented, after a small adjustment suggested by the present reviewer, it has technical-scientific merit compatible with the journal's publications and deserves to be accepted for publication.
The manuscript "Effects of different concentrations of Lactobacillus plantarum and Bacillus licheniformis on silage quality, in vitro fermentation and microbial community of hybrid Pennisetum" is about the effects of Lactobacillus plantarum (LP) and Bacillus licheniformis (BL) on the quality of hybrid Pennisetum (HP) silage. Silages of Pennisetum are widely used in the world for animal nutrition, but the loss in the silos, especially caused by the low low water-soluble carbohydrate is a problem. Thus, the studies about additives aiming to improve the success rate of ensilage are very important to animal nutrition and production systems. The importance of the research field is presented in the introduction section in a very satisfactory way.
There are in the literature some papers about the theme. However, the presented manuscritp, but this one exhibits information about the fermentation and microbial community. Knowledge about that can drive the additive use on forages ensiling.
Adjustment: Inform in full the meaning of "FW" the first time it is mentioned in the text (lines 15 and 77).
The authors could improve the Tables and Figures title, including the meaning of the initials.
Author Response

(The authors gave the same response as above.)

Reviewer 3 Report
Dear Authors,
your paper is interesting, but try to improve the next comments:
Line 17,18, 19 and 21 could be nice and more friendly for the reader to write what does mean, AN, HP, WSC, LA, AA, LP7, DM, CO and ADF
In key word, lactobacillus plantarum and Bacillus licheniformis should be in italics word format.
Ich you put all description of the” letter given in the line 17,18,19 and 21, the you don’t need to describe them in the line 72 to line 82.
Table 1.- maybe put the %-value only with a 1-number after the cero? Additionally, the format of the column from %-value should be homogeneous. After Alfalfa is in another column?
Try to put the table 3 and 5 in one page.
Line 356, 396 please put in italic word, the microorganism’s name
Line 386, what does mean “La”?,
Line 423, a “n” is meaning, and maybe a “,” should be added?.
Line from 451 to 458, if you describe the Bacteria name as genus, please put “sp.” After the name ob them: Lactobacillus sp
Kind regards
Author Response

(The authors gave the same response as above.)
